# Screening of Metabolism-Disrupting Chemicals on Pancreatic α-Cells Using In Vitro Methods

**DOI:** 10.3390/ijms24010231

**Published:** 2022-12-23

**Authors:** Reinaldo Sousa Dos Santos, Ignacio Babiloni-Chust, Laura Marroqui, Angel Nadal

**Affiliations:** 1Instituto de Investigación, Desarrollo e Innovación en Biotecnología Sanitaria de Elche (IDiBE), Universidad Miguel Hernández de Elche, 03202 Elche, Alicante, Spain; 2CIBER de Diabetes y Enfermedades Metabólicas Asociadas, Instituto de Salud Carlos III, Spain

**Keywords:** apoptosis, diabetes, endocrine disruptors, glucagon secretion, metabolism-disrupting chemicals, pancreatic α-cells, test methods

## Abstract

Metabolism-disrupting chemicals (MDCs) are endocrine disruptors with obesogenic and/or diabetogenic action. There is mounting evidence linking exposure to MDCs to increased susceptibility to diabetes. Despite the important role of glucagon in glucose homeostasis, there is little information on the effects of MDCs on α-cells. Furthermore, there are no methods to identify and test MDCs with the potential to alter α-cell viability and function. Here, we used the mouse α-cell line αTC1-9 to evaluate the effects of MDCs on cell viability and glucagon secretion. We tested six chemicals at concentrations within human exposure (from 0.1 pM to 1 µM): bisphenol-A (BPA), tributyltin (TBT), perfluorooctanoic acid (PFOA), triphenylphosphate (TPP), triclosan (TCS), and dichlorodiphenyldichloroethylene (DDE). Using two different approaches, MTT assay and DNA-binding dyes, we observed that BPA and TBT decreased α-cell viability via a mechanism that depends on the activation of estrogen receptors and PPARγ, respectively. These two chemicals induced ROS production, but barely altered the expression of endoplasmic reticulum (ER) stress markers. Although PFOA, TPP, TCS, and DDE did not alter cell viability nor induced ROS generation or ER stress, all four compounds negatively affected glucagon secretion. Our findings suggest that αTC1-9 cells seem to be an appropriate model to test chemicals with metabolism-disrupting activity and that the improvement of the test methods proposed herein could be incorporated into protocols for the screening of diabetogenic MDCs.

## 1. Introduction

Diabetes has been considered one of the most serious metabolic diseases and its global prevalence has been rising at an alarming rate [1]. One of the emerging risks factors that promote diabetes development is our extensive exposure to environmental chemicals that act as endocrine disruptors [2,3].

Despite their differences in etiology and pathogenesis, both type 1 (T1D) and type 2 diabetes (T2D) present dysfunctional glucagon-producing α-cells and impaired glucagon secretion [4,5,6,7]. Glucagon is considered a critical regulator of glycemia because it counteracts the glucose-lowering effects of insulin by stimulating hepatic glycogenolysis and gluconeogenesis [8,9]. In the context of diabetes, dysregulated glucagon secretion may contribute to the observed insulin-induced hypoglycemia in T1D and hyperglycemia in early stages of T2D [10,11].

Accumulating data suggest that metabolism-disrupting chemicals (MDCs), a class of endocrine disruptors that alter the susceptibility to metabolic disorders [12,13], impact β-cell function and survival by different molecular-initiated events (MIEs). For instance, bisphenol A (BPA), a well-studied MDC, can alter insulin secretion in primary mouse islets/β-cells by activating estrogen receptors and altering ion channel expression and activity [14,15,16,17]. Moreover, BPA induces β-cell apoptosis in an estrogen receptor-dependent manner, where BPA increases reactive oxygen species (ROS) generation and disrupts the formation of ERα/ERβ heterodimers [18,19,20]. Tributyltin (TBT), another well-known MDC, induces β-cell dysfunction and demise as well as ROS production by a mechanism involving peroxisome proliferator-activated receptor γ (PPARγ) [20,21,22]. Despite the mounting body of evidence that has been gathered about how MDCs affect β-cells, much less is known regarding the effects of these chemicals on pancreatic α-cells. The fact that α-cell dysfunction poses a risk for both T1D and T2D individuals reinforces the importance of identifying MDCs that may impact α-cell survival and/or function.

As part of GOLIATH, a European Union Horizon 2020-funded project focused on designing MDC testing approaches [23], we have recently described test methods to identify MDCs that may affect β-cells [20]. In our previous study, BPA and TBT were considered as positive controls, while other four chemicals, namely perfluorooctanoic acid (PFOA), triphenylphosphate (TPP), triclosan (TCS), and dichlorodiphenyldichloroethylene (DDE), were used as “unknown” chemicals due to the lack of data about their effects on β-cells. In the present work, we tested these chemicals in α-cells following an adverse outcome pathway framework, where the MIE was studied by pharmacology and two key events, namely ROS production and expression of endoplasmic reticulum (ER) stress markers. As adverse effects, we assessed α-cell viability and glucagon secretion.

## 2. Results

### 2.1. MDCs Affect α-Cell Survival

We have previously shown that the effects of MDCs on β-cell survival could be evaluated by two different approaches, namely the MTT assay and staining with the DNA-binding dyes Hoechst 33342 and propidium iodide (HO/PI) [20]. Here, we used these same methods to evaluate whether different MDCs could affect α-cell viability. For this purpose, the mouse α-cell line αTC1-9 was exposed to a range of concentrations of each MDC. First, we measured cell viability by MTT assay following treatment for 48 h and 72 h (Figure 1A–F and Appendix A). Considering the range of concentrations established based on biomonitoring studies, BPA and TBT decreased α-cell viability in a dose-dependent manner (Figure 1A,B, Appendix A), whereas the other four tested MDCs did not change cell viability (Figure 1C–F, Appendix A). At their highest concentrations, i.e., 1 μM for BPA and 200 nM for TBT, BPA and TBT decreased viability by around 10% (BPA) and 30%, respectively, after treatment for 48 h. Exposure to higher concentrations of PFOA, TPP, TCS, and DDE showed that 1 mM of these chemicals had a robust effect on αTC1-9 cell viability, which was reduced by nearly 80–90% upon treatment (Appendix A).

Next, we assessed αTC1-9 viability using the DNA-binding dyes HO/PI upon 24 h treatment (Figure 1G–I). The high sensitivity of this method allowed us to detect that very low concentrations of either BPA (Figure 1G) or TBT (Figure 1H) induced α-cell apoptosis. While 1 μM BPA promoted a two-fold increase in apoptosis, a three-fold induction was seen in 200 nM TBT-treated cells. As shown by MTT, treatment with concentrations of PFOA, TPP, TCS, and DDE within the range observed in biomonitoring studies did not induce apoptosis (Figure 1I).

As in our previous work [20], a mix of the cytokines interleukin-1β (IL-1β) and interferon-γ (IFNγ) was used as positive control. As expected, IL-1β + IFNγ decreased viability or induced apoptosis in our model (Figure 1 and Appendix A).

### 2.2. Estrogen Receptors and PPARγ Are Involved in BPA- and TBT-Induced α-Cell Apoptosis

BPA is a xenoestrogen that, through the activation of the estrogen receptors ERα, ERβ, and G protein-coupled estrogen receptor (GPER) can affect β-cell function and survival [14,19,20,24]. We observed that BPA-induced apoptosis was totally blunted by concomitant treatment with the pure estrogen receptor antagonist ICI 182,780 (Figure 2A). As TBT is an agonist of both the retinoid X receptors (RXRs) and the PPARγ [25,26], we used the PPARγ antagonist T0070907 to investigate whether PPARγ activation would be involved in TBT-induced apoptosis. As shown in Figure 2B, TBT-induced apoptosis was partially blocked by T0070907. While TBT produced a two- to three-fold increase in apoptosis, treatment with T0070907 blocked 30–40% of TBT-induced apoptosis (Figure 2B). These findings suggest that estrogen receptors and PPARγ are involved in the MIE whereby BPA and TBT, respectively, induce α-cell death.

### 2.3. BPA and TBT Promote ROS Generation 

Previous data show that exposure to BPA and TBT, but not to PFOA, stimulate ROS production in β-cells [18,19,20,21,22]. Hence, we measured ROS generation in α-cells upon exposure to all six MDCs. In αTC1-9 cells, a 30–40% increase in ROS levels was observed upon treatment with BPA or TBT, whereas PFOA, TPP, TCS, and DDE did not change ROS production in these cells (Figure 3A–F). Of note, menadione was used as a positive control.

### 2.4. Exposure to Different MDCs Does Not Induce ER Stress

Accumulating evidence suggests that some endocrine disruptors induce ER stress, which may lead to apoptosis in a variety of models [27]. We measured the protein expression of two ER stress markers, namely immunoglobulin heavy chain-binding protein (BiP) and the phosphorylated form of the eukaryotic initiation factor-2α (p-eIF2α), in αTC1-9 cells treated with each of the six chemicals for 24 h. Exposure to 1 μM BPA reduced BiP expression by 45%, while no changes were observed in p-eIF2α levels (Figure 4A–C). Treatment with TBT slightly augmented BiP levels without changing p-eIF2α expression (Figure 4D–F). Expression of BiP and p-eIF2α was not modified by exposure to PFOA, TPP, TCS, and DDE (Figure 4G–I).

### 2.5. α-Cell Function Is Perturbed by Different MDCs

As a test method to assess α-cell function, we measured static glucagon secretion under stimulatory and inhibitory conditions (0.5 mM and 11 mM glucose, respectively) upon exposure to different doses of each MDC for 48 h (Figure 5). BPA (Figure 5A) and TBT (Figure 5B) did not change glucose-regulated glucagon secretion, despite a trend in cells treated with higher concentrations of TBT. The greatest changes were observed in cells exposed to PFOA; at a stimulatory glucose concentration (i.e., 0.5 mM), PFOA induced a decrease in glucagon secretion in a dose-dependent manner, reaching a reduction of nearly 25% at 1 μM compared with vehicle-treated cells. At high glucose, 10 pM PFOA augmented glucagon secretion (Figure 5C). Exposure to TPP also diminished glucagon release at low glucose, albeit only 100 pM TPP was statistically significant (Figure 5D). Curiously, we observed that only 10 nM TCS and 100 pM DDE significantly reduced glucose-regulated glucagon secretion at a stimulatory glucose concentration, while exposure to higher doses of each MDC did not change secretion (Figure 5E,F). Treatment with 1 μM TCS also increased glucagon release at high glucose (Figure 5E). Of note, a 30–40% decrease in glucagon secretion upon incubation at 11 mM glucose observed in our experiments is in line with previous studies [28,29].

## 3. Discussion

Aiming to detect environmental pollutants that could have metabolism-disrupting activity in α-cells, here we propose different in vitro test methods to assess α-cell viability (MTT assay and staining with DNA-binding dyes) and function (glucagon secretion).

Up until now, there are no human α-cell lines available for research use as they are difficult to generate, mainly due to the paucity of information about the glucagon promoter and its lack of specificity for α-cells [30]. Among the limited α-cell lines existing for rodents, we used the αTC1 Clone 9 (or αTC1-9), which is more differentiated than the parental αTC1 cell line and produces only glucagon (but not insulin or preproinsulin) [31]. Since its cloning from the parental αTC1 cells, studies by us and others have confirmed that αTC1-9 cells are a useful model to investigate α-cell physiology. This cell line allows us to assess several cellular parameters, including Ca^2+^ signaling [29,32], electrical activity [32], and glucagon secretion [28,29,33,34,35]. Furthermore, αTC1-9 response to different stimuli, such as proinflammatory cytokines and leptin, is similar to their primary cell counterparts [31,32,36,37].

### 3.1. α-Cell Viability Tests

Using two previously established approaches to determine cell viability, namely the MTT assay and HO/PI DNA-binding dyes [20], we showed that low doses of BPA and TBT induced α-cell apoptosis, while only extremely high doses (i.e., >100 μM) of PFOA, TPP, TCS, and DDE negatively affected α-cell viability.

Although recent evidence shows that BPA affects β-cell survival [18,19,20,38,39], BPA effects on α-cells are still unclear. To our knowledge, there is only a recently published study that analyzed pancreatic sections from rats given 4.5 µg/L BPA in drinking water for 45 days [40]. In this study, α-cells from BPA-treated rats presented signs of apoptosis, such as cell shrinkage and small dense nuclei, as well as higher levels of caspase 3 reactivity compared to vehicle-treated animals. Glucagon immunostaining confirmed that BPA-treated rats had less α-cells than the control rats [40]. Despite the different approaches, these data agree with our results showing that BPA induces apoptosis in vitro.

BPA is a known xenoestrogen that induces β-cell apoptosis via activation of the estrogen receptors ERα, ERβ, and GPER [19,20]. As all three estrogen receptors are expressed in α-cells [41], we investigated whether these receptors were involved in BPA-induced apoptosis in αTC1-9 cells. In line with our previous findings in β-cells, the pure estrogen receptor antagonist ICI 182,780, which can bind to and block both ERα and ERβ [42], prevented BPA-induced apoptosis, suggesting that the activation of estrogen receptors is part of the MIE underlying BPA effects in α-cells. 

It has been reported that in vivo and in vitro exposure to TBT results in β-cell death [21,22,43,44]. Here, we show for the first time that TBT also induces α-cell apoptosis at doses as low as 1 nM. Curiously, at the highest dose, the TBT effect on viability was stronger than the mix of proinflammatory cytokines used as positive control. This effect on viability was partly abrogated by a PPARγ antagonist, which suggests that this receptor is involved in TBT-induced apoptosis, as we have previously described in β-cells [20]. As TBT can also bind to and activate RXR [25,26], it seems likely that this receptor may be also involved in TBT-induced α-cell apoptosis. RXR activation may explain why a PPARγ antagonist did not completely block TBT effects. Thus, further studies are needed to evaluate RXR involvement in TBT-induced changes in α-cells. PFOA, TPP, TCS, and DDE only reduced cell viability when αTC1-9 cells were exposed to concentrations higher than 100 µM, which agrees with prior studies in β-cells. We showed that concentrations of PFOA above 20 µM induced apoptosis in the human EndoC-βH1 cell line and the rat INS-1E cells [20]. In the rat RIN-m5F β-cell line, doses of PFOA higher than 100 µM induced apoptosis after 48 h, whereas 1 µM PFOA did not affect cell survival [45]. In MIN6, a mouse β-cell line, as well as in primary mouse islets, up to 300 µM PFOA did not change viability; doses above 500 µM, however, induced apoptosis upon 24 h treatment [46]. Up to 1 µM TPP did not induce apoptosis in EndoC-βH1 and INS-1E cells as well as in dispersed mouse islets, even after 72 h exposure [20]. Exposure to 17 and 35 μM TCS increased necrosis in MIN6 cells, while lower doses (up to 7 μM) did not change viability in MIN6, EndoC-βH1, and INS-1E cells [20,47]. Lastly, DDE treatment ranging from 10 fM to 50 µM had no effect on the viability of different β-cell lines, namely EndoC-βH1, INS-1E, and MIN6, or in dispersed mouse islets, when cells were treated for up to 48 h [20,48,49]. On the other hand, longer DDE exposure (from 4 to 8 days) diminished viability even at the lowest dose in INS-1E cells [49]. Taken together, our findings suggest that αTC1-9 cells are a reliable model for the identification of MDCs that could affect α-cell survival. Moreover, we confirm that the test methods used herein are suitable for the assessment of cell viability/apoptosis in response to MDCs.

Following an adverse outcome pathway framework, we next investigated two key events that could be involved in cell death and dysfunction, namely oxidative stress and ER stress [50,51]. For this purpose, we measured ROS production and the expression of two ER stress markers, namely BiP and p-eIF2α. As it has been previously reported in β-cells [18,19,20,21,22,38], BPA and TBT induced ROS production in αTC1-9 cells; of note, exposure to the other four MDCs did not change ROS production at the concentrations tested. As ROS levels were increased only by chemicals that induced apoptosis, it is reasonable to state that the generation of ROS is a key event in the pathway to MDC-induced α-cell apoptosis. Regarding the ER stress markers, the expression of the chaperone BiP was reduced by 1 µM BPA and slightly increased by 200 nM TBT. PFOA, TPP, TCS, and DDE did not alter BiP expression. None of the six chemicals tested changed the expression of the activated form of eIF2α (i.e., p-eIF2α) at the doses tested. At first glance, this lack of major changes in the expression of ER stress markers may be unexpected for BPA and TBT, as these MDCs, which induce α-cell apoptosis, activate the ER stress response in other cell types [27,52,53,54,55,56]. However, it is important to keep in mind that the expression of ER stress markers may depend on the amount and time of exposure to a given stimulus. Moreover, it is possible that the activation of the ER stress response is not part of the mechanism whereby BPA and TBT induce apoptosis in α-cells. Finally, contrary to β-cells, where ER stress is a key component contributing to β-cell loss, α-cells are more resistant to ER stress-mediated apoptosis [57]. Overall, these data suggest that MDC-induced ROS production is a key event that may be used as a substitute for measurement of cell viability, as it properly predicted α-cell apoptosis.

### 3.2. α-Cell Function Tests

As glucagon is a hormone responsible for opposing the hypoglycemic effects of insulin, the maintenance of adequate α-cell function is crucial for the regulation of glucose homeostasis.

Even though the effects of different MDCs on insulin release have been largely explored [12,58,59], whether exposure to these chemicals affects α-cell function remains largely unknown. The only study available to date showed that acute treatment with 1 nM BPA suppressed low glucose-induced intracellular Ca^2+^ oscillations in primary mouse α-cells within intact pancreatic islets [60]. As intracellular Ca^2+^ signaling is intimately linked to glucagon secretion (see below), one might speculate that BPA-induced suppression of Ca^2+^ oscillations could ultimately inhibit glucagon release at low glucose concentrations. Unfortunately, glucagon secretion was not measured in Alonso-Magdalena et al.’s study. In the present work, we observed that glucose-regulated glucagon secretion remained unchanged upon BPA exposure for 48 h. It would be interesting to investigate whether Ca^2+^ signals are altered in the αTC1-9 cell line.

Contrary to β-cells, where TBT increases glucose-stimulated insulin secretion in a variety of models [20,21,22,44], here we did not observe a clear effect on glucagon secretion. However, as a clear trend is seen at both glucose concentrations, it may be that higher TBT doses result in augmented glucagon release. 

The four MDCs that did not affect α-cell viability, namely PFOA, TPP, TCS, and DDE, disturbed α-cell function in some way. Among them, PFOA was the one that induced the most changes in glucagon secretion, where doses ranging from 100 pM to 1 µM PFOA impaired glucagon secretion at glucose-stimulating levels. Our in vitro findings seem to contradict a recent in vivo study wherein the oral administration of PFOA to adult male mice (1.25 mg/kg PFOA for 28 days) resulted in high blood glucagon levels, increased fasting glycemia, and augmented hepatic gluconeogenesis [61]. These apparently opposing outcomes may be due to different PFOA doses and exposure times between studies. The concentrations of PFOA used herein are within the range observed in biomonitoring studies, which varied between 650 pM and 34 nM [62]. Zheng et al., on the other hand, observed that the serum levels of PFOA at the end of exposure were around 134 µM [61]. Furthermore, in the present study, we treated αTC1-9 cells for only 48 h, while Zheng and collaborators exposed mice to PFOA for 28 days, which may be long enough to allow α-cells to adapt and start secreting more glucagon. Unfortunately, as we have not tested longer time points, we cannot directly compare our findings with the aforementioned in vivo study. Along with prior studies showing that PFOA exposure also impairs β-cell function [20,46,63], our findings in α-cells strongly suggest that PFOA acts as a diabetogenic MDC.

Interestingly, only single concentrations of either TCS (10 nM) or DDE (100 pM) affected glucagon secretion at a stimulatory glucose concentration. As these findings may have important implications for risk assessment toxicology, it will be interesting to further explore the mechanisms underlying TCS and DDE effects on α-cell function.

Contrary to β-cells, where the stimulus-secretion coupling model is well-established, the exact mechanisms regulating glucose-modulated glucagon secretion are still under debate. Glucagon secretion can be regulated by intrinsic (exerted within the α-cell itself) and paracrine (involvement of factors released from neighboring β- and δ-cells) mechanisms. A generally accepted intrinsic model suggests that, at low glucose concentrations, there is less glucose entering the cells via the glucose transporters, which will result in decreased intracellular ATP/ADP ratio. This lower ATP/ADP ratio leads to moderate activity of K_ATP_ channels, which causes membrane depolarization and subsequent opening of voltage-dependent Na^+^ and Ca^2+^ channels. The resulting increase in intracellular Ca^2+^ triggers exocytosis of glucagon secretory granules. Activation of store-operated Ca^2+^ entry due to ER Ca^2+^ depletion and cAMP-induced potentiation of Ca^2+^-induced Ca^2+^ release from the ER also contribute to the exocytosis of glucagon-containing vesicles [8,9,64,65]. Our findings suggest that MDCs affecting α-cell function could be disrupting any step of this complex secretory pathway. In fact, considering our experimental approach, i.e., direct treatment of a cell line, it is likely that PFOA, TPP, TCS, and DDE may be interfering with intrinsic mechanisms controlling glucagon release, such as glucose transport/metabolism, activity/expression of ion channels, and/or exocytosis. Additionally, as we used an α-cell line instead of whole pancreatic islets, we cannot discard the possibility that any of the MDCs tested in our study may affect the paracrine regulation of glucagon secretion. This option should be investigated in future studies. Either way, our study shows that MDC-induced α-cell dysregulation may contribute to the diabetogenic actions of these chemical pollutants.

In conclusion, our present findings suggest that αTC1-9 cells represent a valid model for the identification of MDCs with potential diabetogenic activity. We also validated in α-cells the test methods for the assessment of cell viability presented in our previous study. The generation of ROS, but not the expression of ER stress markers, could be a suitable surrogate for the evaluation of cell viability. Lastly, the measurement of glucagon secretion seems to be a valuable test method to explore whether a given MDC affects α-cell function.

## 4. Materials and Methods

### 4.1. Chemicals

Chemicals used in this work were acquired as follows: BPA (Cat. No. 239658), TBT (Cat. No. T50202), PFOA (Cat. No. 77262), TPP (Cat. No. 241288), TCS (Cat. No. PHR1338), and DDE (Cat. No. 123897) were obtained from Sigma-Aldrich (Barcelona, Spain). MDC stock solutions were weekly prepared by dissolution in 100% cell-culture grade, sterile-filtered DMSO (Sigma-Aldrich; Cat No D2650) and stored at −20 °C between uses. Recombinant human IL-1β was obtained from R&D Systems (Abingdon, UK; Cat. No. 201-LB/CF) and recombinant murine IFNγ from PeproTech (Rocky Hill, NJ, USA; Cat. No. 315-05). ICI 182,780 (Cat. No. 1047) and T0070907 (Cat. No. HY-13202) were obtained from Tocris Cookson Ltd. (Avonmouth, UK) and MedChem Express (Monmouth Junction, NJ, USA), respectively. T0070907 was redosed every 8–10 h due to its short half-life.

### 4.2. Culture of αTC1-9 Cells

Mouse glucagon-releasing cell line αTC1-9 (RRID: CVCL_0150, ATCC, Manassas, VA, USA) was cultured in DMEM (Invitrogen, Barcelona, Spain) without phenol red supplemented with 2 mM l-glutamine, 19 mM NaHCO_3_, 15 mM HEPES, 10% inactivated FBS, 100 U/mL penicillin, 0.1 mg/mL streptomycin, 0.1 mM non-essential amino acids, and a final glucose concentration of 16 mM [35]; of note, this glucose concentration has been routinely used to culture this cell line [29,31,32,36]. Cells were kept at 37 °C in 95% humidified air and 5% CO_2_.

### 4.3. Assessment of Cell Viability by MTT Assay

The MTT assay was performed as previously described [20]. Briefly, MTT was added to each well (final concentration: 0.5 mg/mL) and incubated for 3 h at 37 °C. Following incubation, the supernatant was removed by aspiration and formazan crystals were dissolved by addition of 100 µL DMSO. The absorbance was measured at 595 nm using an iMark™ Microplate Absorbance Reader (Bio-Rad, Hercules, CA, USA) and the percentage of cell viability was calculated.

### 4.4. Assessment of Cell Viability by DNA-Binding Dyes

Percentage of living, apoptotic and necrotic cells was assessed upon staining with DNA-binding dyes Hoechst 33342 and propidium iodide [66,67]. A minimum of 500 cells per experimental condition was counted by two different observers, one of them being unaware of sample identity to avoid bias (agreement between results from both observers was >90%). Results are expressed as percentage of apoptosis.

### 4.5. DCF Assay

The generation of ROS was evaluated using the fluorescent probe 2′,7′-dichlorofluorescein di-acetate (DCF; Sigma-Aldrich, Barcelona, Spain) as described in [20]. DCF fluorescence was quantified in a POLASTAR plate reader (BMG Labtech, Ortenberg, Germany). Data are represented as DCF fluorescence corrected by total protein. Menadione (15 μM for 90 min) was employed as a positive control.

### 4.6. Western Blotting

Cells were washed with cold PBS and lysed in Laemmli buffer. Cells extracts were resolved on 12% SDS-PAGE and transferred to nitrocellulose membrane (Bio-Rad, Hercules, CA, USA). Immunoblot analysis was performed by overnight incubation with antibodies against BiP, p-eIF2α, and α-Tubulin as described in [19]. The antibodies used herein are listed in Appendix A.

### 4.7. Glucagon Secretion

Cells were washed and then preincubated in a modified Krebs-Ringer buffer (120 mM NaCl, 5.4 mM KCl, 1.2 mM KH_2_PO_4_, 1.2 mM MgSO_4_, 2.4 mM CaCl_2_, 20 mM HEPES pH 7.4, and 0.1% bovine serum albumin) to which 5.6 mM glucose was added for 2 h before stimulation. At the end of this incubation, cells were sequentially stimulated with low (0.5 mM) and then high glucose (11 mM) for 30 min (each stimulation). After each stimulatory period, the incubation medium was collected, placed onto ice, and centrifuged at 1400× *g*, 5 min at 4 °C. The supernatant was transferred into a fresh tube containing aprotinin (20 mg/L) and stored at −80 °C until glucagon measurements. Glucagon levels were assessed using a mouse glucagon ELISA kit (Mercodia, Uppsala, Sweden). The amount of glucagon released was first normalized by total protein and then by glucagon secretion at low glucose in vehicle-treated cells.

### 4.8. Data Analysis

The GraphPad Prism 7.0 software (GraphPad Software, La Jolla, CA, USA) was used for statistical analyses. Data are presented as the mean ± SEM. Statistical analyses were performed using Student’s *t*-test, one-way ANOVA or two-way ANOVA as stated in the figure legends. One- and two-way ANOVA were followed by Dunnett’s test as post hoc analysis. *p* values ≤ 0.05 were considered statistically significant.

## Figures and Tables

**Figure 1 ijms-24-00231-f001:**
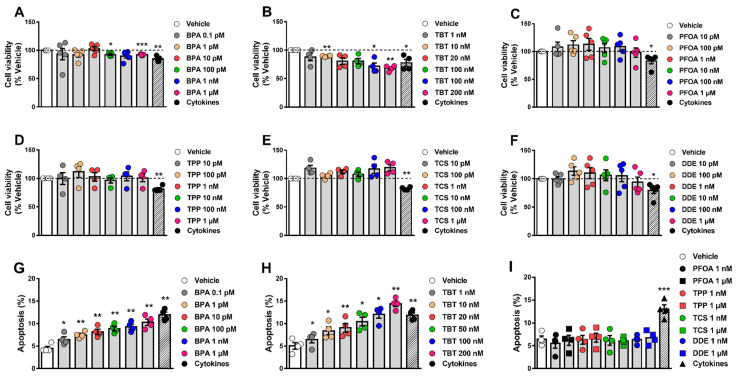
α-cell viability upon MDC exposure. (**A**–**F**) αTC1-9 cells were treated with vehicle (DMSO) or different doses of BPA (**A**), TBT (**B**), PFOA (**C**), TPP (**D**), TCS (**E**), or DDE (**F**) for 48 h. A cocktail of the cytokines IL-1β + IFNγ (50 and 1000 U/mL, respectively) was used as a positive control. Cell viability was evaluated by MTT assay. (**G**–**I**) αTC1-9 cells were treated with vehicle (DMSO) or different doses of BPA (**G**), TBT (**H**), PFOA, TPP, TCS, or DDE (**I**) for 24 h. Apoptosis was evaluated using HO and PI staining. Data are shown as means ± SEM (n = 4–5 independent experiments, where each dot represents an independent experiment). * *p* ≤ 0.05, ** *p* ≤ 0.01 and *** *p* ≤ 0.001 vs. Vehicle. MDCs vs. Vehicle by one-way ANOVA; Cytokines vs. Vehicle by two-tailed Student’s *t* test.

**Figure 2 ijms-24-00231-f002:**
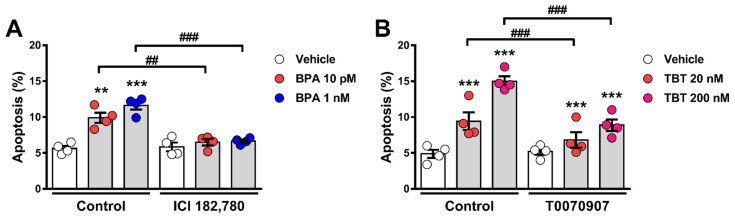
Estrogen receptors and PPARγ are implicated in BPA- and TBT-induced α-cell apoptosis, respectively. (**A**) αTC1-9 cells were treated with vehicle (DMSO) or BPA (10 pM or 1 nM) in the absence (control) or presence of 1 µM ICI 182,780 for 24 h. (**B**) αTC1-9 cells were treated with vehicle (DMSO) or TBT (20 nM or 200 nM) in the absence (control) or presence of 100 nM T0070907 for 24 h. Apoptosis was evaluated using HO and PI staining. Data are shown as means ± SEM (n = 4 independent experiments, where each dot represents an independent experiment). ** *p* ≤ 0.01 and *** *p* ≤ 0.001 vs. its respective vehicle; ^##^
*p* ≤ 0.01 and ^###^
*p* ≤ 0.001 as indicated by bars. Two-way ANOVA.

**Figure 3 ijms-24-00231-f003:**
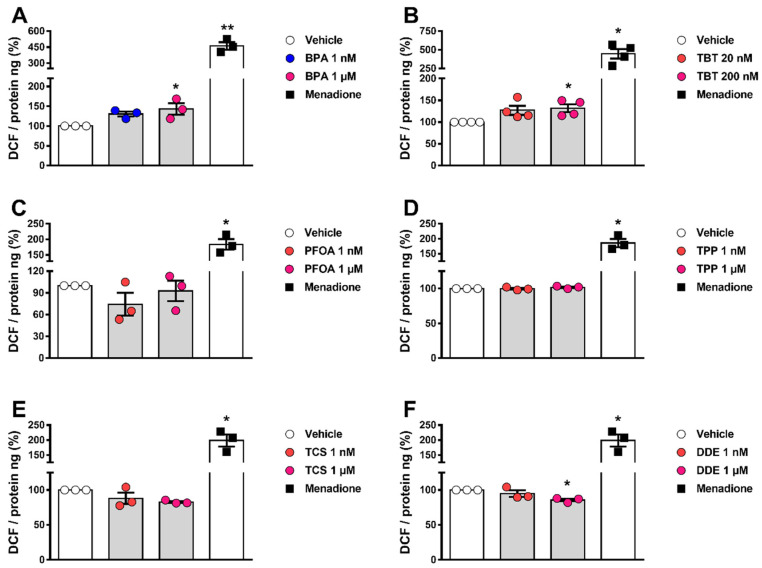
ROS production upon MDC exposure. αTC1-9 cells were treated with vehicle (DMSO) or different doses of BPA (**A**), TBT (**B**), PFOA (**C**), TPP (**D**), TCS (**E**), or DDE (**F**) for 24 h. Menadione (15 μM for 90 min) was used as a positive control. Generation of ROS was assessed by oxidation of the fluorescent probe DCF and normalized by total protein. Data are shown as means ± SEM (n = 3–4 independent experiments, where each dot represents an independent experiment). * *p* ≤ 0.05 and ** *p* ≤ 0.01 vs. vehicle. MDCs vs. vehicle by one-way ANOVA; menadione vs. vehicle by two-tailed Student’s *t* test.

**Figure 4 ijms-24-00231-f004:**
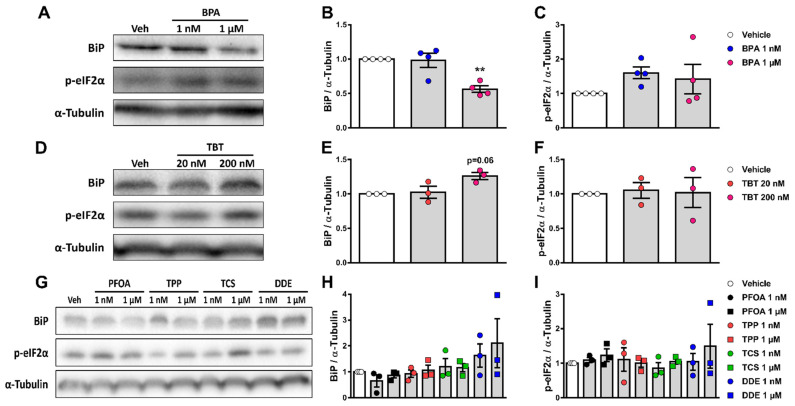
Expression of ER stress markers upon MDC exposure. αTC1-9 cells were treated with vehicle (DMSO) or different doses of BPA (**A**–**C**), TBT (**D**–**F**), PFOA, TPP, TCS, or DDE (**G**–**I**) for 24 h. Protein expression was measured by Western blot. Representative images of three independent experiments are shown (**A**,**D**,**G**) and densitometry results are presented for BiP (**B**,**E**,**H**) and p-eIF2α (**C**,**F**,**I**). Data are shown as means ± SEM (n = 3–4 independent experiments, where each dot represents an independent experiment). ** *p* ≤ 0.01 vs. vehicle. MDCs vs. vehicle by one-way ANOVA.

**Figure 5 ijms-24-00231-f005:**
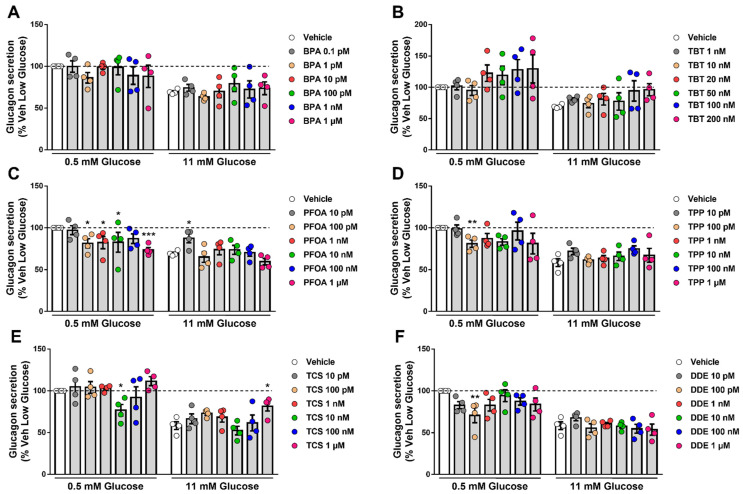
Glucagon secretion upon MDC exposure. αTC1-9 cells were treated with vehicle (DMSO) or different doses of BPA (**A**), TBT (**B**), PFOA (**C**), TPP (**D**), TCS (**E**), or DDE (**F**) for 48 h. Glucagon secretion was measured at 0.5 and 11 mM glucose, and glucagon released into the medium was measured by ELISA. Data are normalized to glucagon secretion at low glucose (i.e., 0.5 mM) in vehicle-treated cells. Data are shown as means ± SEM (n = 4 independent experiments, where each dot represents an independent experiment). * *p* ≤ 0.05, ** *p* ≤ 0.01 and *** *p* ≤ 0.001 vs. its respective vehicle. Two-way ANOVA.

## Data Availability

Not applicable.

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
