# Peer review of "Screening of Metabolism-Disrupting Chemicals on Pancreatic α-Cells Using In Vitro Methods"

_ijms, 2022, doi:10.3390/ijms24010231_

Round 1
Reviewer 1 Report
This manuscript describes the testing of perfluorooctanoic acid, triphenylphosphate, triclosan and dichlorophenyldichloroethylene in alpha-cells following an adverse outcome pathway framework. In performed experiments, the molecular initiated events were studied by pharmacology, ROS production and expression of endoplasmic reticulum stress markers. The Authors used wide range of methods including MTT assay, DCF assay, Western blotting and glucagon secretion. The manuscript is well written. Obtained results are clearly presented on numerous figures and they are discussed in the separate section. In my opinion, the manuscript can be accepted for publication in Int. J. Mol. Sci. after the following issues are addressed: 3.1 and 3.2 section names should start with the capital letter, in section 2.5 the symbol alpha is missing.
Author Response
We are grateful to the Reviewer for the positive comments and for believing
our manuscript can be accepted for publication.
We have addressed the issues pointed out by the Reviewer in the revised version of the manuscript.
Reviewer 2 Report
Although the overall idea of the manuscript "Screening of metabolism-disrupting chemicals on pancreatic α-cells using in vitro methods" is good, there are some important questions and I'd say problems with this manuscript.
One of the main problems is the very high (16 mM) concentration of glucose used for cell culturing. Obviously it's a diabetic concentration, not normal concentration. Why don't the authors use the classical glucose concentrations? Why do they also treat cells with so low, unphysiological (0.5 mM) glucose in one of the experiments?
This problem leads to a question why there is so little difference in glucagon production between the 0.5 mM and 11 mM glucose levels. A possible answer is that the cell culture was grown at 16 mM glucose and the pre-incubation with an intermediate glucose concentration doesn't help. There is no real value of glucagon secretion provided except for 100%, but do the authors have a calibration to evaluate this and provide the real values? Right now it seems, the cells are stressed by the diabetic conditions at which they were grown, and the evaluation of MDC (endocrine disruptors with obesogenic and/or diabetogenic action) in such conditions is weird.
Rather late addition of aprotinin and absence of other protease inhibitors may also be an important drawback in the glucagon assay protocol.
The 95% O2 atmosphere also corresponds to hyperoxia, which should lead to oxygen toxicity.
Other questions and remarks are listed below.
Paragraph 2.2. The Fig.2 data support the importance of estrogen receptors. Could you clarify better, which of them are blocked by the used ligand? As for the PPARg (Fig. 2B), the significances, shown on the graph, don't really support the statement. Some strict comparisons like "two-fold" or another should be added to the text. Still, evaluation of retinoid X receptors (RXRs), mentioned in the paragraph, is suggested by the data. You should address this point in the text too.
Remarks to the figures:
Figure 2: use of the word "vehicle" for two different things (DMSO) or absence of a chemical is misleading. Please correct the figure and corresponding text to any appropriate way of presentation.
Figure 1: The figures of "apoptosis", should be presented in a better agreement with the MTT data, so that the idea could be more evident. Right now the figure (I) lacks the data from two drugs, presented separately in (G, H). Such difference in presentation may lead to unnecessary questions.
Figure 5: It looks suspicious, that a 15(?)% difference with so few data points is significant in (C). Please check the statistics in all the graphs. You should also add a dash line (100% or vehicle) to all graphs and figures to make the differences easier to compare.
If there is no mistakes in statistics, please provide the data of Fig. 5C as unpublished material, at the next round of the revision.
Mistakes such as "2.5.(-. cell function is perturbed by different MDCs" in one of the paragraphs titles further support the need of a careful checkup of the data, figures and text by the authors.
A remark to the figures providing the results of different concentrations of the drugs:
The data of Fig 5E and 5F with statistical differences of just one intermediate concentration of a chemical is suspicious too, even though the authors try to convince the readers with references [63,64] and a brief discussion. Such result in this study might be a consequence of a pipetting or dilution error or any other one... Could you decipher what is meant by the phrase "n = 4-5 independent experiments, where each dot represents an independent experiment" in the figure legends? How many technical repeats and independent growth experiments were used etc., so that it was understandable from the text.
Generally, the use of linear graphs showing the effect vs concentration and a regression curve is preferable for such data.
If all the questions to the results are answered, I'd recommend adding a scheme to the discussion, describing the most important result(s) and the outcome(s). The separate "Conclusion" section would also look better in this text.
Author Response
We are grateful to the Reviewer for the comments, and much appreciate his/her thorough revision and suggestions for improvement.
Please see the attachment for our point by point rebuttal letter.

Round 2
Reviewer 2 Report
The authors significantly improved the revised version of the manuscript. Both most important points (glucose and O2 concentrations) were addressed either as a clarification on common practices in the field or as a mistake correction, respectively. The calculations of O2 concentration are also interesting and could lead to an interesting discussion, but as the used O2 concentration is normal, there is no need in further arguments. I'm sure, one could obtain an interesting and significant effect of 95% O2 compared to 20%, despite the difference in concentration for the cells may be not very high.
The authors also addressed most of my minor comments, including the ones to the figures. As for the Fig.1, I'd stand at my suggestion to improve the presentation of data, but the current view is also fine.
The provided statistics and data, as well as ATCC information sheet give a good impression and provide all the needed information. Although it's always important to note that the glucose concentrations (either 16 mM or 0.5 mM) are not physiological, they are in line with the commonly used protocols applied to these cells.
The paper certainly can be published in this special issue of IJMS devoted to MDC screening and testing.